# KD-64—A new selective A$_{2A}$ adenosine receptor antagonist has anti-inflammatory activity but contrary to the non-selective antagonist—Caffeine does not reduce diet-induced obesity in mice

**Magdalena Kotańska**[1☯]*, **Anna Dziubina**[2☯], **Małgorzata Szafarz**[3☯], **Kamil Mika**[1☯], **Karolina Reguła**[1☯], **Marek Bednarski**[1☯], **Małgorzata Zygmunt**[1☯], **Anna Drabczyńska**[4☯], **Jacek Sapa**[1☯], **Katarzyna Kieć-Kononowicz**[4☯]

**1** Department of Pharmacological Screening, Jagiellonian University Medical College, Krakow, Poland, **2** Department of Pharmacodynamics, Jagiellonian University Medical College, Krakow, Poland, **3** Department of Pharmacokinetics and Physical Pharmacy, Jagiellonian University Medical College, Krakow, Poland, **4** Department of Technology and Biotechnology of Drugs, Faculty of Pharmacy, Jagiellonian University Medical College, Krakow, Poland

☯ These authors contributed equally to this work.
* magda.dudek@uj.edu.pl

**Data Availability Statement:** All relevant data are within the manuscript.

## Abstract

The A$_2$ adenosine receptors play an important role, among others, in the regulation of inflammatory process and glucose homeostasis in diabetes and obesity. Thus, the presented project evaluated of influence of the selective antagonist of A$_{2A}$ adenosine receptor—KD-64 as compared to the known non-selective antagonist—caffeine on these two particular processes. Two different inflammation models were induced namely local and systemic inflammation. Obesity was induced in mice by high-fat diet and the tested compounds (KD-64 and caffeine) were administrated for 21 days. KD-64 showed anti-inflammatory effect in both tested inflammation models and administered at the same dose as ketoprofen exerted stronger effect than this reference compound. Elevated levels of IL-6 and TNF-α observed in obese control mice were significantly lowered by the administration of KD-64 and were similar to the values observed in control non-obese mice. Interestingly, caffeine increased the levels of these parameters. In contrast to caffeine which had no influence on AlaT activity, KD-64 administration significantly lowered AlaT activity in the obese mice. Although, contrary to caffeine, KD-64 did not reduce diet-induced obesity in mice, it improved glucose tolerance. Thus, the activity of the selective adenosine A$_{2A}$ receptor antagonist was quite different from that of the non-selective.

## Introduction

Obesity is defined as over-storage of lipids in adipose tissue that occurs when the amount of supplied energy significantly exceeds its consumption by the body [1]. Currently, it is an

**Funding:** KK-K received each award. This work was supported by statutory funds N42/DBS/000039 from the Faculty of Pharmacy Jagiellonian University Medical College, Krakow, Poland. The funders had no role in study design, data collection and analysis, decision to publish, or preparation of the manuscript.

**Competing interests:** The authors have declared that no competing interests exist.

extremely important civilization problem, since obesity is considered a risk factor for cardio-vascular diseases (e.g. ischemic heart disease, hypertension, heart failure, stroke), diabetes, dyslipidemia, autoimmune diseases and even cancer [2]. The reasons for overweight (and obesity) are numerous and varied. Genetic, environmental and/or social factors as well as the hormonal status of the body play the major role in its pathogenesis [3]. According to the current theory, development of obesity, associated with adipocyte hypertrophy and hyperplasia, is connected not only with disturbances in the secretory function of the fat tissue, but also with increased inflammatory activation of adipocytes, release of pro-inflammatory cytokines and dysregulation of adipokine secretion [4, 5, 6, 7]. An increase in the levels of pro-inflammatory cytokines and proteins such as interleukin-6 (IL-6), tumor necrosis factor alpha (TNF-α), leptin or C-reactive protein (CRP) in both blood and adipose tissue itself was observed as obesity progressed. On the other hand, reduced levels of anti-inflammatory factors for instance adiponectin [8] as well as various changes (increase or decrease) in IL-10 levels together with the mostly unchanged levels of IL-8 [9, 10] seem to be also relevant. Until now, the mechanism triggering inflammatory activation of adipose tissue has not been clearly defined. Different theories point to the role of hypoxia of adipocytes [11], cellular stress during obesity development [12], and elevated glucose level, all generating large amounts of free oxygen radicals in adipocytes and stimulating pro-inflammatory cytokines secretion [13, 14].

Pharmacological treatment of obesity remains an unresolved problem. The role of adenosine receptor signalling in the development and progression of numerous diseases has been emphasized for years [15]. Adenosine is an endogenous purine nucleoside that participates in the development of obesity [16]. It works through adenosine receptors such as A$_1$, A$_{2A}$, A$_{2B}$ and A$_3$ that differ in their pharmacological profile (including affinity for adenosine), tissue localization and the system of second order messengers. Adenosine receptors are widely expressed in organs and tissues involved in metabolism regulation such as liver, pancreas, adipose tissue and muscles, moreover its presence on immune cells [17], points to their significant role in the inflammatory processes. Since all of these receptors are engaged in glucose homeostasis, adipogenesis, insulin resistance, inflammation and thermogenesis, treatment with specific agonist and/or antagonists could normalize several mechanisms involved in pathophysiology of obesity [18].

The A$_{2A}$ receptor being the most abundant adenosine receptor in human and murine white and brown adipose tissue, as well as on immune cells and organs has become a potential target for obesity studies [17, 19]. Data indicate that adenosine signalling via A$_{2A}$ is required for activation of brown adipose tissue and protects mice from diet-induced obesity [20]. It has been also demonstrated that A$_{2A}$ receptor knockout mice exhibit impaired thermogenesis, oxygen consumption, and lipolysis [21]. Moreover, Csóka et al., (2017) observed the reduced food intake in such mice and consequently a lower body mass as compared to control animals [22]. A strong functional interaction between the dopamine D$_2$ and adenosine A$_{2A}$ receptors [23] has been further discovered and a blockade of adenosine A$_{2A}$ receptor can even mimic the action of D$_2$ agonists [24]. Since dopamine is known to be an important regulator of energy expenditure [25, 26] and food intake [27] reduced dopamine signal transduction may give rise to overeating and decreased energy consumption, both of which contribute to the positive energy balance seen in obesity [28].

The adenosine A$_{2A}$ receptor is predominantly expressed on inflammatory cells, including neutrophils, mast cells, macrophages, monocytes, and platelets [29] and in many animal studies it has been demonstrated that its activation reduced inflammatory processes [29, 30] and improved molecular markers of inflammation. On the other hand, it has been shown that the direct local injection of the selective A$_{2A}$ receptor antagonist ZM 241385 (from the group of non xanthine adenosine receptors antagonists) in carrageenan-induced inflammatory

**Fig 1. Chemical scheme of annelated xanthine derivatives (A), structure of compound KD-64 (B), structure of caffeine (C).**

hyperalgesia reduced inflammatory hypersensitivity suggesting that activation of peripheral adenosine A$_{2A}$ receptors during inflammation is associated with mechanical hyperalgesia [31]. Zygmunt et al., (2015) presented the series of A$_{2A}$ adenosine receptor ligands, a group of arylalkyl pyrimido [2,1-f]-purinediones (Fig 1A) with significant anti-inflammatory activity in carrageenan-induced paw edema model. These compounds—annelated xanthine derivatives —have similar structure to KD-64 –ligand (Fig 1B) we have used in the presented study [32].

For comparison purposes caffeine (1,3,7-trimethyl xanthine) (Fig 1C) was chosen, a nonselective A$_1$ and A$_{2A}$ adenosine antagonist and the most popular and well-studied methylxanthine which has been reported as thermogenic and lipolysis stimulator leading to fat oxidation in adipocytes and release of glycerol and fatty acids to the bloodstream [33, 34]. Moreover, caffeine modulates glucose metabolism and increases energy expenditure, as well as has impact on [35] body fat reduction [36] and weight loss [37] mostly as adjuwant agent [38, 39] especially during physical exercise or when administered simultaneously with the calorie restriction diets [40, 41].

While non-selective adenosine A$_1$ and A$_{2A}$ receptor antagonists such as caffeine are well investigated in obesity-related mechanisms, the results of studies on selective A$_{2A}$ receptor antagonists are still inconsistent, therefore we have chosen for our experiments a selective A$_{2A}$ receptor antagonist, designated as KD-64, with significant selectivity over other adenosine receptors (K$_i$ [μM] values are 0.24, > 25, > 10 and > 10 for A$_{2A}$, A$_1$, A$_{2B}$ and A$_3$ receptors, respectively) [42].

In the first part of our research we have estimated the effect of KD-64 on the inflammation in the carrageenan or zymosan induced models of inflammation. Further, in a diet-induced

mice obesity model, we have confirmed the development of inflammation, and compared the effects of investigated compound and caffeine, i.e. selective and non-selective $A_{2A}$ receptor antagonists, on the primary metabolic variables.

## Materials and methods

### Animals

Adult (six-week old) male Albino Swiss mice, CD-1, weighing 25–30 g were used in the inflammatory models and estimation of locomotor activity and adult (six-week old) female Albino Swiss mice, CD-1, weighing 19–22 g were used in the model of obesity. The animals were obtained from the Animal House of the Faculty of Pharmacy of the Jagiellonian University Medical College. Animals were kept in environmentally controlled rooms, in standard cages lit by an artificial light for 12h each day. They had free access to food and water, except for the time of the acute experiment. The randomly established experimental groups consisted of 8 mice. All animal care and experimental procedures were carried out in accordance with European Union and Polish legislation acts concerning animal experimentation, and were approved by the Local Ethics Committee at the Jagiellonian University in Cracow, Poland (Permission No: 256/2015 and 55/2017).

During the experiments, when possible, steps were taken to minimize animal suffering and distress. Unfortunately, in the anti-inflammatory studies, it was not possible to administer any anaesthetics and analgesics since they would interfere with the tests, however the animals were not isolated individually to avoid developing stress. In the studies of anti-obesity properties efforts were made to create the safe and stress-free conditions for animals so that stress would not affect the results of experiments. During the induction of obesity, only two people responsible for the subsequent administration of the investigated compounds, were allowed to handle the animals. After experiment mice were killed by decapitation that is considered to be the most humane method of mouse euthanasia.

### Drugs, chemical reagents and other materials

Ketoprofen was used as standard anti-inflammatory compound and was purchased from Sigma-Aldrich (Poland). Carrageen was purchased from FCM Corporation (USA), Zymozan A and Evans blue from Sigma-Aldrich (Poland), Caffeine (used as standard in obese model) was purchased from Alfa-Aesar (Poland). KD-64—(1,3-dimethyl-9-((1*r*,4*r*)-4-methylcyclohexyl-6,7,8,9-tetrahydropyrimido[2,1-*f*]purine-2,3(1*H*,3*H*)-dione) (Fig 1B) [43] was synthesized in the Department of Technology and Biotechnology of Drugs, Faculty of Pharmacy, Jagiellonian University Medical College, Cracow, Poland. NMR and LC-MS techniques assessed identity and purity of final product. In all experiments ketoprofen, caffeine or KD-64 were administered as suspensions in 1% Tween 80 in sterile water for injection.

### Inflammation models

**Carrageenan-induced edema model.** To induce inflammation, 0.1 ml of 1% carrageenan solution in water was injected into the hind paw subplantar tissue of mice, according to the modified method of C. A. Winter and P. Lence [44, 45], as described previously [46]. The development of paw edema was measured with a plethysmometr (Plethysmometr 7140, Ugo Basile). Prior to the administration of the tested substances (KD-64 or ketoprofen), paw diameters were measured and data were recorded for further comparison. The KD-64 compound was administered at the doses of 1, 5 or 10 mg/kg, intraperitoneal (*ip*), prior to carrageenan injection, similarly ketoprofen (reference standard) was administered at the dose of 5 mg/kg

[47]. 1% Tween 80 (vehicle) was administered by the same route to the control group (it had no effect on edema). Results were presented as changes in the hind paw volume 3 h after carrageenan administration. Immediately, after measurement mice were administrated 2500 units/ mice of heparin *ip* and 20 minutes later sacrificed by decapitation. Blood was collected to the Eppendorf tubes and centrifuged at 600 x g (15 min, 4˚C) in order to obtain plasma used for the determination of CRP levels.

**Zymosan A induced peritoneal inflammation.**   Peritoneal inflammation was induced as described previously [48]. Zymosan A was freshly prepared (2 mg/ml) in sterile 0.9% NaCl and 30 minutes after *ip* injection of the investigated compounds (KD-64 or ketoprofen at the dose of 5 mg/kg), zymosan A was injected via the same route. Four hours later the animals were killed by decapitation and blood was collected into a heparin-containing tubes. After centrifugation mice plasma was stored for the further measurements of the CRP levels. The peritoneal cavity was lavaged with 1.5 ml of PBS and after 30s of gentle manual massaging the exudates were retrieved. Cells were counted using an optical microscope (DM1000, Leica) and Bürker hemocytometer following staining with Turk's solution.

**Vascular permeability.**   In the control studies Evans blue was suspended in the saline (10 mg/ml) and injected intravenously (*iv*) into the caudal vein, which was immediately followed by *ip* injection of zymosan A. Thirty minutes later the animals were killed by decapitation and their peritoneal cavities were lavaged with 1.5 ml of saline as described above. The lavage fluid was centrifuged and the absorbance of the supernatant was measured at 620 nm in order to assess vascular permeability by peritoneal leakage of *iv* injected Evans blue as described previously [48]. In the subsequent experiments KD-64, ketoprofen (reference standard) both at the dose of 5 mg/kg or vehicle (administered *sc*, control group) were injected *ip* 30 min before Evans blue and zymosan A. Further procedures were the same as in the control group.

## Locomotor activity

The locomotor activity was recorded with an Opto M3 multichannel activity monitor (Multi-Device Software v1.3, Columbus Instruments, USA). It was evaluated as the distance travelled by the animals while attempting to climb upward [49]. After *ip* administration of KD-64 compound at the doses of 1, 5 or 10 mg/kg, each mouse was placed in a cage for a 30 minutes' habituation period. After that time the number of crossings of photo beams was measured for 20 minutes.

## Obesity study

**Metabolic disturbance induced with a high-fat/sucrose diet and its influence on body weight and spontaneous activity.**   Mice were fed on high-fat diet consisting of 40% fat blend (Labofeed B with 40% lard, Morawski Feed Manufacturer, Poland) for 15 weeks, water and 30% sucrose solution were available *ad libitum* [50, 51]. Control mice were fed on a standard diet (Labofeed B, Morawski Feed Manufacturer, Poland) and drank water only. After 12 weeks, mice with obesity were randomly divided into three equal groups that had the same mean body weight and were treated *ip* with tested compounds at the following doses: KD-64 5 mg/kg b.w./day or caffeine 50 mg/kg b.w./day and control group: 1% Tween 80 (vehicle) 0.35 ml/kg (fat/sugar diet + vehicle = obesity control group) once daily between 9:00 and 10:00 AM for 21 days. Control mice (control without obesity) were maintained on a standard diet, with *ip* administration of 1% Tween 80, 0.35 ml/kg (standard diet + vehicle = control group). Animals always had free access to feed, water and sucrose. After experiment mice were killed by decapitation and plasma was harvested to determine levels of TNF-α, IL-6 and activity of alanine aminotransferase (AlaT).

High-fat feed composition: protein–193 g, fat (lard)–408 g, fiber–28.1 g, crude ash–43.6 g, calcium–9.43 g, phosphorus–5.99 g, sodium–1.76 g, sugar–76 g, magnesium–1.72 g, potassium–7.62 g, manganese–48.7 mg, iodine–216 mg, copper–10.8 mg, iron–125 mg, zinc–61.3 mg, cobalt–0.253 mg, selenium–0.304 mg, vitamin A–15000 units, vitamin D3–1000 units, vitamin E–95.3 mg, vitamin K3–3.0 mg, vitamin B1–8.06 mg, vitamin B2–6.47 mg, vitamin B6–10.3 mg, vitamin B12–0.051 mg, folic acid–2.05 mg, nicotinic acid–73.8 mg, pantothenic acid–19.4 mg, choline–1578 mg, lysine 9.0 g, methionine + cysteine 6.3 g, tryptophan 2.0 g, threonine 6.0 g, isoleucine 6.0 g, leucine 12.0 g, valine 8.0 g, histidine 4.0 g, arginine 10.0 g, phenylalanine 7.0 g, tyrosine 5.5 g, betaine 17.0 g.

Standard feed composition: protein–175 g, fat–35 g, fiber–70 g, crude ash–32 g, starch–330 g, calcium–9.5 g, phosphorus–7.5 g, magnesium–3 g, potassium–7.5 g, sodium–1.9 g, manganese–50 mg, iodine–144 mg, sulfur–1,9 g, zinc–50 mg, copper–11 mg, iodine–200 mg, selenium–0.4 mg, vitamin A–12000 units, vitamin D3–800 units, vitamin E–78 mg, vitamin K3–2.4 mg, vitamin B1–8 mg, vitamin B2–7 mg, vitamin B6–11 mg, vitamin B12–0.042 mg, folic acid–25 mg, nicotinic acid–94 mg, pantothenic acid–25 mg, choline–1900 mg, lysine 9.0 g, methionine + cysteine 6.3 g, tryptophan 2.0 g, threonine 6.0 g, isoleucine 6.0 g, leucine 12.0 g, valine 8.0 g, histidine 4.0 g, arginine 10.0 g, phenylalanine 7.0 g, tyrosine 5.5 g, betaine 17.0 g.

The high-fat diet contained 550 kcal and the standard diet 280 kcal per 100 g.

The spontaneous activity of mice was measured on the 1$^{st}$ and 21$^{st}$ day of the treatment with a special RFID-system—TraffiCage (TSE-Systems, Germany). The animals were subcutaneously implanted with a radio-frequency identificator (RFID), which enabled to count the presence and time spent in different areas of the cage. The obtained data was grouped using an appropriate computer program [49].

## Biochemical analysis

**Glucose tolerance test.** The glucose tolerance test was performed at the beginning of 16$^{th}$ week. After twenty administrations of the tested compounds (KD-64 or caffeine), food and sucrose were discontinued for 20h and then glucose tolerance was tested. Glucose (1g/kg b.w.) was administrated *ip* [50, 51] and blood samples were taken from the tail vein at the time points: 0 (before glucose administration), 30, 60 and 120 minutes after administration. Glucose levels were measured with glucometer (ContourTS, Bayer, Germany, test stripes: ContourTS, Ascensia Diabetes care Poland Sp. z o.o., Poland, REF:84239666). The area under the curve (AUC) was calculated using the trapezoidal rule.

**Insulin tolerance test.** Insulin tolerance was tested on the next day after the glucose tolerance test. Mice had free access to standard food and water, but 3h before insulin tolerance test the food was taken away. Insulin (0.5 IU/kg b.w.) was injected *ip* and blood samples were collected at the time points: 0, 15 and 30 minutes from the tail vein and glucose levels were measured with glucometer (ContourTS, Bayer, Germany, test stripes: ContourTS, Ascensia Diabetes care Poland Sp. z o.o., Poland, REF:84239666) [50, 51]. The AUC was calculated using the trapezoidal rule.

**Plasma levels of IL-6, TNF-α, CRP and AlaT activity.** On the next day after insulin tolerance test, 20 minutes after *ip* administration of heparin (2500 units/mice) animals were sacrificed by decapitation. The blood was collected and then centrifuged at 600 x g (15 min, 4˚C) in order to obtain plasma. To determine AlaT activity in the plasma samples, standard enzymatic spectrophotometric test (Biomaxima S.A. Lublin, Poland, catalogue number: 1-023-0150) was used. In order to quantify IL-6 and TNF-α LANCE® Ultra Detection Kits (PerkinElmer, Inc, USA, catalogue numbers: TRF1505, TRF1504C/TRF1504M) were used. For the

determination of CRP standard enzymatic spectrophotometric tests (Shanghai Sunred Biological Technology Co., Ltd, China, catalogue number: 201-02-0219) were applied.

## Statistical analysis

The obtained results were analyzed using a one-way variance analysis (ANOVA), followed by a Dunnett post-hoc test, with the significance level set at 0.05 (locomotor activity, AlaT activity, IL-6 or TNF-α levels,), a two-way variance analysis (ANOVA), followed by a Bonferroni post-hoc test (changes in body weight) or a Multi-t test (glucose tolerance test, insulin tolerance test, spontaneous activity). The results were expressed as the means ± standard error of the mean (SEM). Graph Pad Prism 6.0 was used for data analysis.

## Results

### Influence of KD-64 on locomotor activity

Compound KD-64 did not affect spontaneous activity in mice after single intraperitoneal administration at all tested doses. The results are shown in Fig 2.

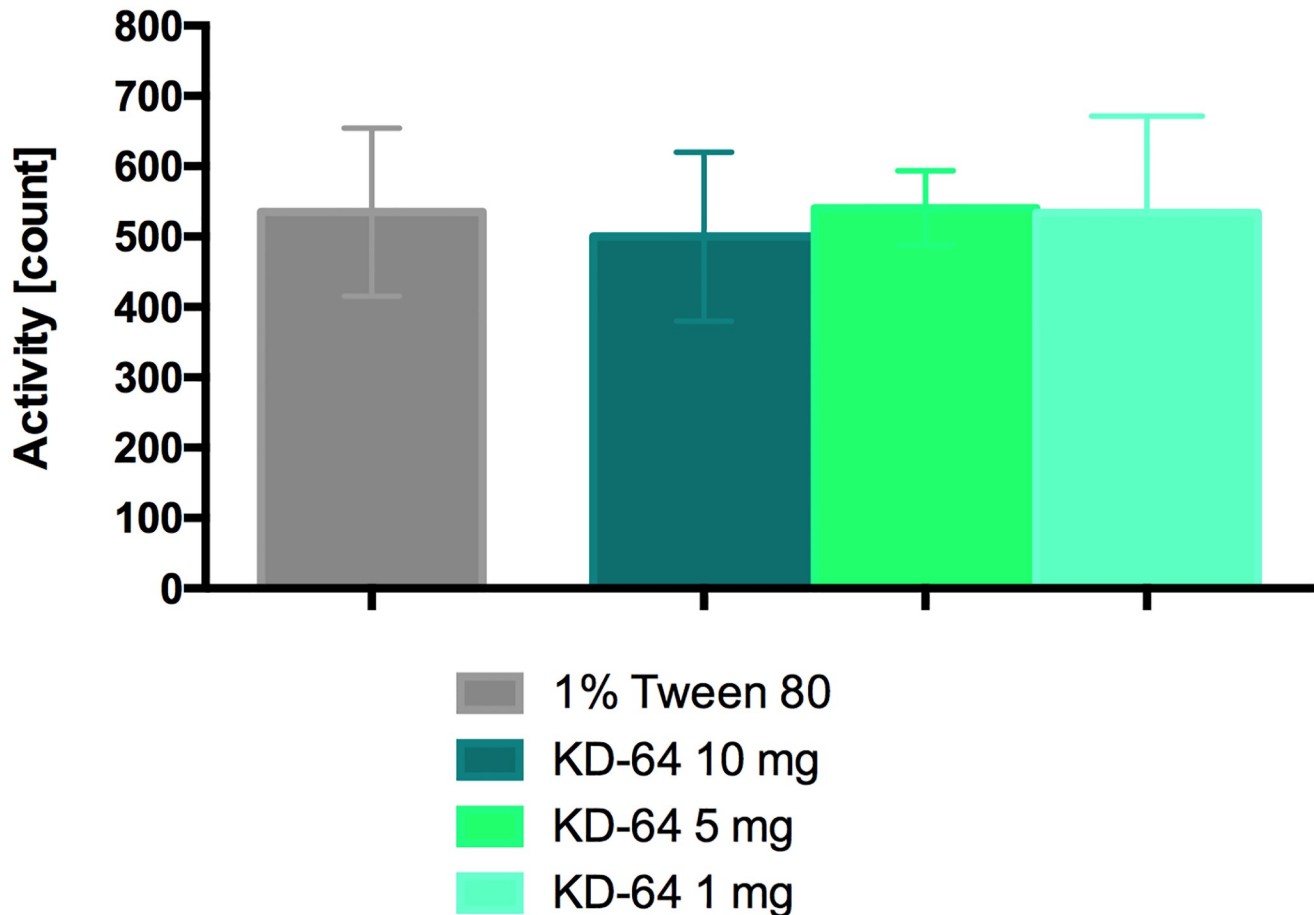

**Fig 2. Locomotor activity after a single administration of KD-64.** Results are mean ± SEM, n = 6. Comparisons were performed using one-way ANOVA Dunnet's post hoc test.

## Influence of KD-64 on carrageenan-induced paw edema in mice

The mouse paw became edematous after the injection of carrageenan, and in the control group edema reached a peak at 3h (increase by 97.7% of the initial volume). The increase in paw edema was significantly inhibited by the KD-64 administration at a dose of 5 mg/kg b.w. as compared to the control group, which was given carrageenan and vehicle only (Fig 3A). Since the results were comparable to the ones observed in the group receiving ketoprofen, 5 mg/kg b.w. of KD-64 (as an active dose) has been selected for further studies. In the plasma of mice treated with KD-64 significant decrease in CRP level (similar to the one observed after ketoprofen administration) was also determined (Fig 3B).

## Influence of KD-64 on zymosan A induced peritoneal inflammation and vascular permeability

The early infiltration of neutrophils measured at 4h after zymosan-induced peritonitis was significantly inhibited in the group receiving KD-64 at the dose of 5 mg/ kg b.w. as compared to the control group, which was given zymosan A alone and leucocytosis was comparable to the one measured after ketoprofen administration (Fig 4A). CRP concentration in mice plasma was also decreased in the group receiving KD-64, interestingly levels of CRP after ketoprofen administration did not change significantly (Fig 4B). The early vascular permeability measured at 30 min after zymosan-induced peritonitis was significantly inhibited in the group receiving KD-64 compared to the control group, which was given zymosan A alone (Fig 4C). After ketoprofen administration vascular permeability also decreased however it was still significantly higher than in control group without induced peritoneal inflammation. Thus, in both experiments KD-64 administered at the same dose as ketoprofen exerted stronger effect than reference compound.

## Influence of KD-64 on body weight and peritoneal fat

Mice fed with high-fat/sugar diet showed more weight gain throughout the 12-week period of inducing obesity as compared to the control group. Animals fed with high-fat diet and treated with KD-64 at the dose of 5 mg/kg b.w. showed significantly less weight gain than mice from the obese control group, however only during the first week of administration. From the second week of KD-64 administration the difference wasn't significant. Mice from the group receiving caffeine (50 mg/kg b.w./day, *ip*) starting from the first week of treatment gained less weight compared to the control group and at the end of the experiment weighed significantly less. The results are shown in Fig 5A and 5B. Animals consuming high-fat feed had also significantly higher amount of fat in peritonea. The results are shown in Fig 5C.

## Influence of KD-64 on plasma IL-6 and TNF-α levels in obese mice

In obese control mice higher plasma levels of IL-6 and TNF-α were observed than in control standard fed mice. However, they were significantly lowered by the administration of KD-64 for 21 days at the dose of 5 mg/kg b.w./day and were similar to the values observed in control non-obese mice (Fig 6). Interestingly, caffeine increased the levels of these parameters, and they were significantly higher not only vs. levels in standard fed control group but also vs. obese control group.

## Influence of KD-64 on AlaT activity in obese mice

Activity of AlaT in plasma of obese mice was significantly higher than in standard diet fed control mice. Administration of caffeine had no influence on AlaT activity, surprisingly KD-64 at the tested dose of 5 mg/kg significantly lowered AlaT levels in obese mice (Fig 7).

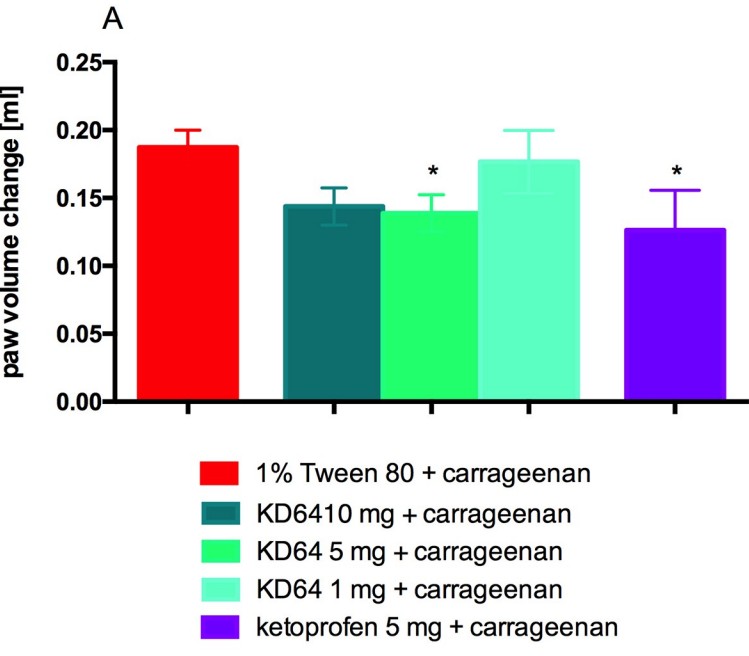

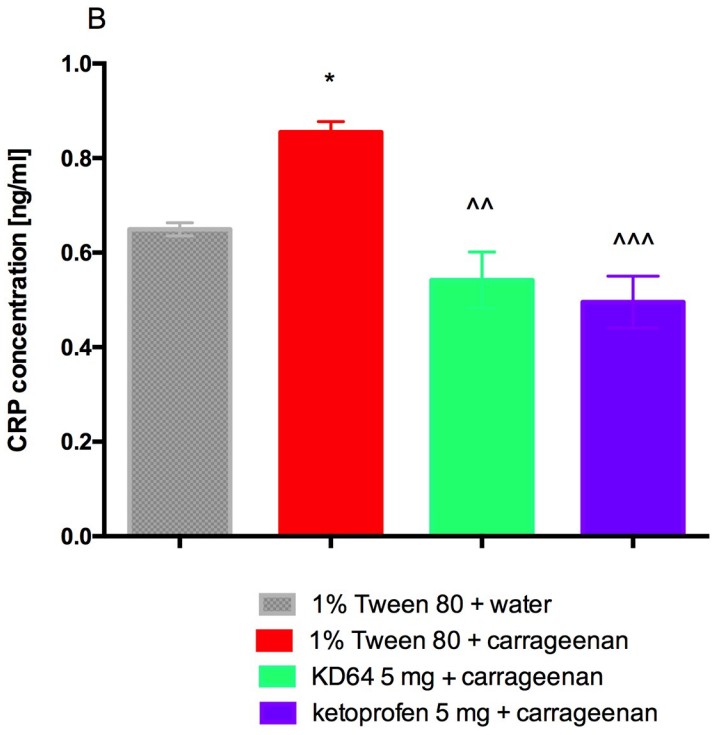

**Fig 3. Anti-inflammatory effects of the tested compounds in the carrageenan-induced paw edema test.** (**A**) Changes in the paw volume in 3h after drug administration in relation to the initial volume (before carrageenan injection). Results are mean ± SEM, n = 8. Comparisons were performed using one-way ANOVA Dunnet's post hoc test. * Significant against control mice administered carrageenen; *p<0.05. (**B**) Concentration of C-reactive protein in plasma. Results are mean ± SEM, n = 8. Comparisons were performed by t-Student test. * Significant against control mice, ^ Significant against control mice administered carrageenen; *p<0.05, ^^p<0.01, ^^^p<0.001.

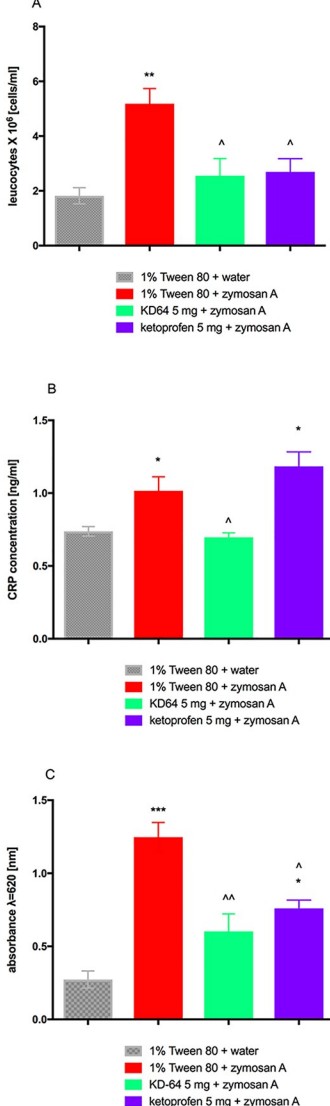

**Fig 4. Anti-inflammatory effects of the tested compounds in model of zymosan-induced peritonitis in mice.** (**A**) Neutrophil infiltration during zymosan-induced peritonitis in mice, (**B**) Concentration of C-reactive protein in plasma, (**C**) Vascular permeability. Results are mean ± SEM, n = 8. Comparisons were performed by t-Student test. * Significant against control mice, ^ Significant against control mice administered zymosan. *,^p<0.05, **,^^p<0.01, ***p<0.001.

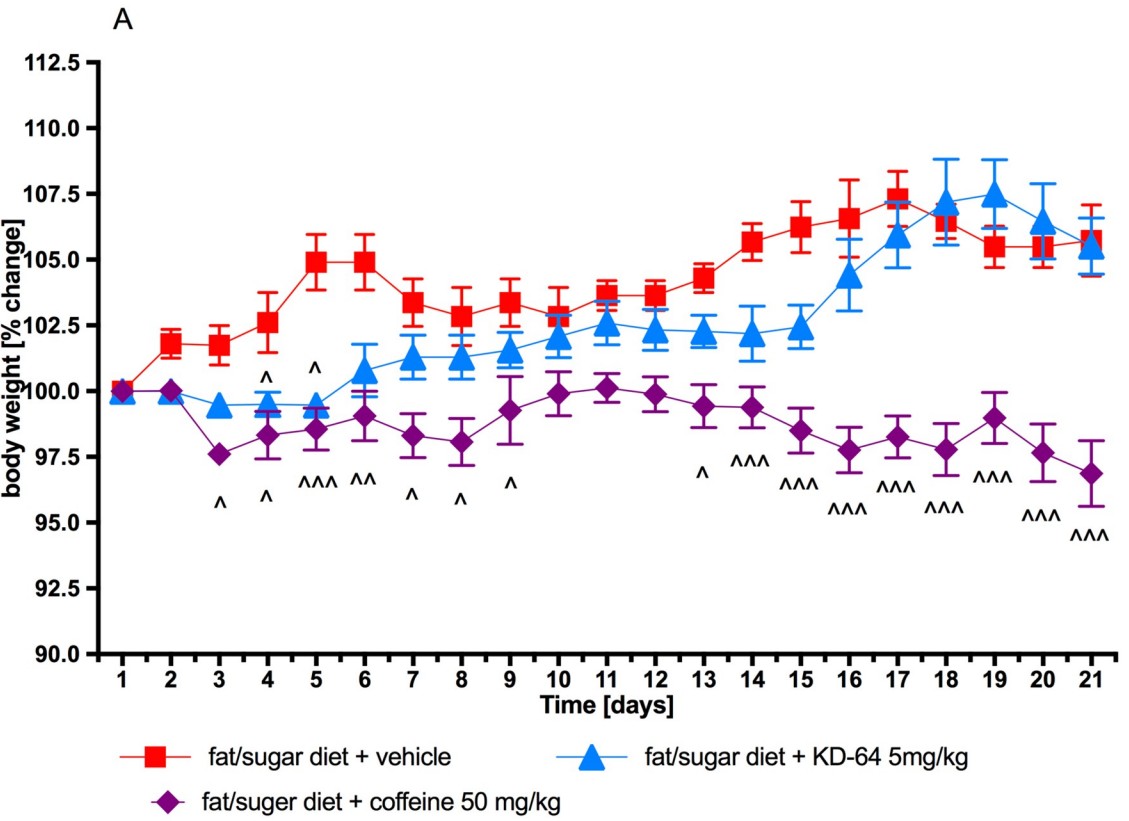

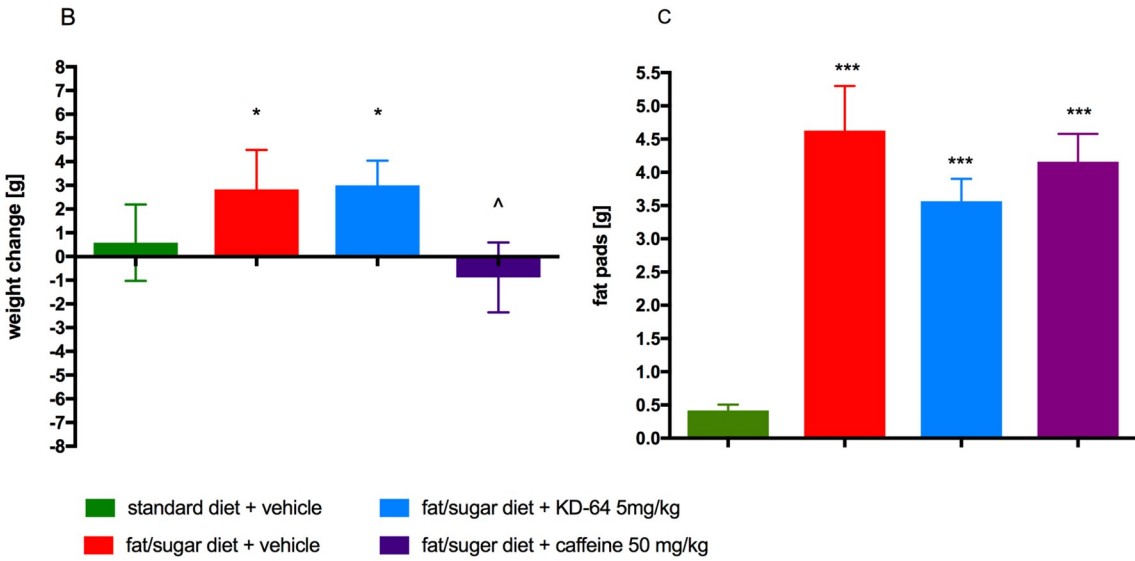

**Fig 5. Effect of administration of KD-64 or caffeine on body weight and mass of adipose pads.** (**A**) Percent change of body weight during the administration. (**B**) Sum of weight changes. (**C**) Mass of adipose pads. Results are expressed as means ± SEM, n = 8. Multiple comparisons were performed by two-way ANOVA, Bonferroni's post hoc (**A**) or one-way ANOVA Dunnet's post hoc tests (**B**, **C**). ^ Significant against control mice fed fat/sugar diet; * Significant against control mice fed standard diet; ^p<0.05, ^^p<0.01, ***, ^^^p<0.001.

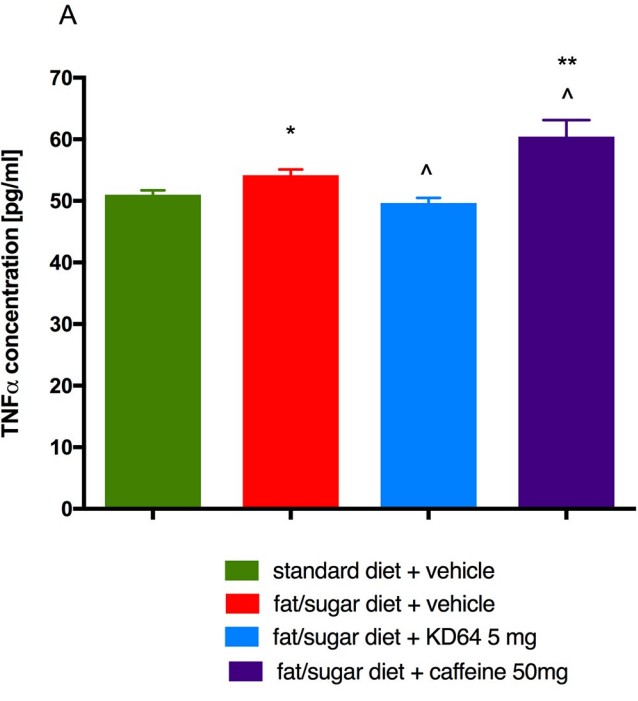

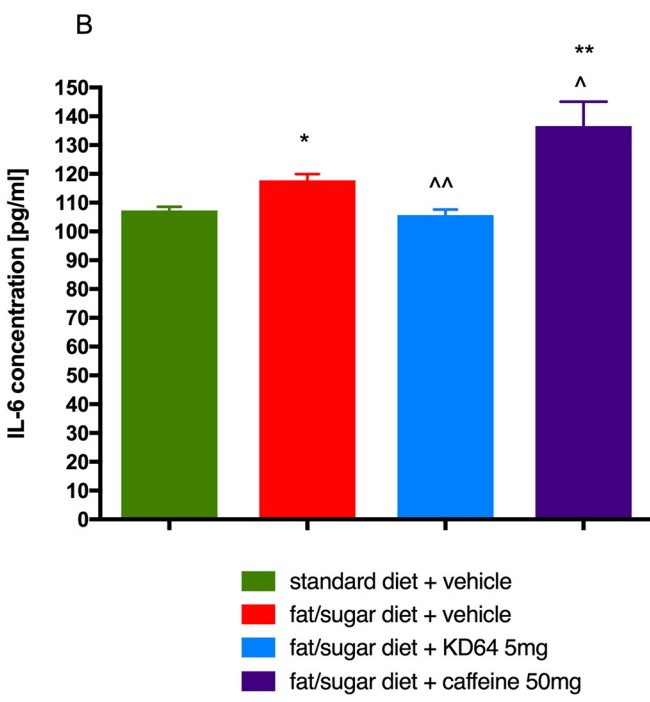

**Fig 6. Effect of administration of KD-64 or caffeine on TNF-α (A) and IL-6 (B) levels in plasma.** Results are expressed as means ± SEM, n = 8. Comparisons were performed by one-way ANOVA Dunnet's post hoc test. * Significant against control mice fed standard diet; ^ Significant against control mice fed fat/sugar diet; *,^ p<0.05, **, ^^p<0.01.

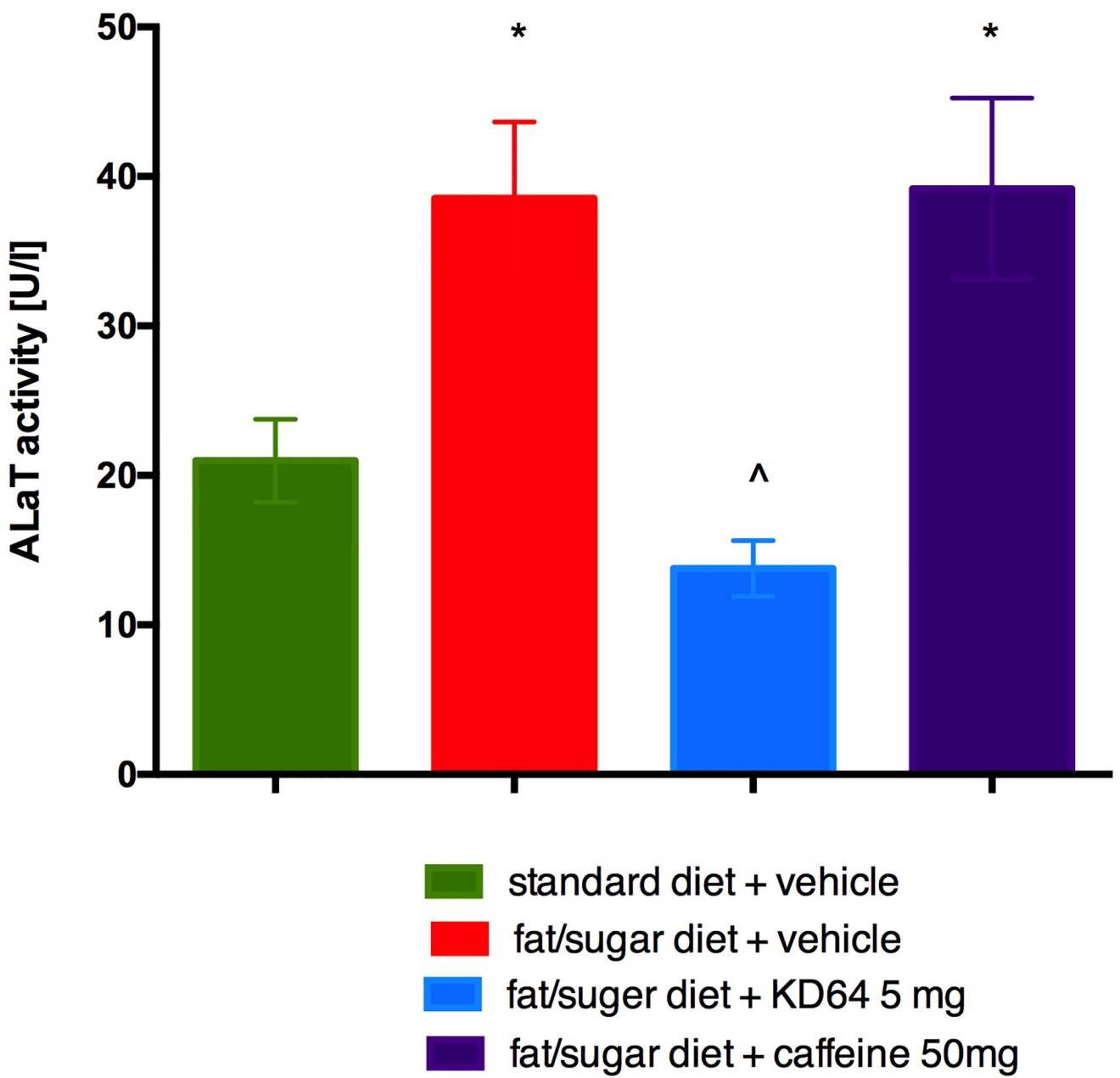

**Fig 7. Effect of administration of KD-64 or caffeine on alanine aminotransferase activity in plasma.** Results are expressed as means ± SEM, n = 8. Comparisons were performed by one-way ANOVA Dunnet's post hoc test. * Significant against control mice fed standard diet, ^ Significant against control mice fed fat/sugar diet; *,^ p<0.05.

## Glucose tolerance and insulin sensitivity after KD-64 treatment of obese mice

At 30 minutes after glucose load the blood glucose levels of mice in all tested groups receiving high fat diet were similar and significantly higher as compared to the levels determined in control standard diet fed mice. At the subsequent time points (60 and 120 minutes after glucose load) there was no statistical difference observed between glucose blood levels of standard diet

fed mice and high fat diet fed mice treated with KD-64 or caffeine. Interestingly, at the last time point (120 minutes after glucose load) glucose levels determined only in mice treated with KD-64 compound were similar to the levels observed in control standard diet fed mice and significantly lower than in control obese mice. Results are shown in Fig 8A. At the same time, as shown in Fig 8B, the AUC was decreased by KD-64 treatment at the dose of 5 mg/kg b.w. as compared to both obese control group and group treated with caffeine, however it was still higher than in control standard diet fed group.

In the insulin test, neither KD-64 nor caffeine affected blood glucose levels, which were similar in all tested groups (Fig 9).

## Influence of KD-64 on spontaneous activity

Compound KD-64 at the tested dose did not affect spontaneous activity in obese mice after a single *ip* administration, but spontaneous activity decreased during certain hours after twentieth administration vs. spontaneous activity in control group. Caffeine, on the other hand, increased spontaneous activity during certain hours after both first and twentieth *ip* administration of the tested dose. The results are shown in Fig 10.

## Discussion

The $A_2$ adenosine receptors play an important role in regulation of glucose homeostasis in both diabetes and obesity, but they also take active part in the inflammatory processes and these two particular abilities of $A_2$ adenosine receptors were a subject of the presented study. In the obesity model we have tested activity of the new selective antagonist of $A_{2A}$ adenosine receptor—compound KD-64 and compared its effect to the known non-selective antagonist of adenosine receptors—caffeine. Subsequently in two different inflammation models the activity of KD-64 has been compared to the activity of potent anti-inflammatory agent—ketoprofen.

From the available literature it is known that caffeine—non-selective adenosine $A_{2A}$ receptor antagonist is able to inhibit various obesity-related abnormalities, including low metabolism, adiposity, dyslipidemia, systemic/tissue inflammation, and insulin resistance [52]. Thus, we began to wonder whether the selective adenosine $A_{2A}$ receptor antagonist may have similar properties—especially when it comes to inflammation and obesity?

For research, the selective $A_{2A}$ adenosine receptor antagonist KD-64 has been chosen, which in previous studies, administered at the dose of 5 mg/kg b.w., exerted antiparkinsonian activity [43]. Preliminary experiments aimed at the determination of spontaneous and anti-inflammatory activities after a single administration of the tested compound. The purpose of this study was to select the lowest dose of KD-64 compound that has anti-inflammatory activity, but simultaneously does not influence spontaneous activity so it can be administered chronically in the obesity model. In obesity studies it is particularly important that the tested compounds do not increase activity, which could further contribute to an increase in energy consumption and an undesirable effect of the psyche—agitation. On the other hand, it is also important that the tested compounds do not reduce activity, because sedation, for example, may cause a decrease in food intake and consequently weight loss that could be attributed as a non-specific effect. Such effect will also be unacceptable in chronic therapy since chronic fasting can be harmful to the body. Therefore, for the safe and effective compounds tested in obesity models and with potential action towards reducing body weight, it is crucial to have no effect on spontaneous activity [53]. In the case of presented studies, it is especially important since adenosine, through $A_1$ and $A_{2A}$ receptors, is involved in the regulation of spontaneous activity [54, 55]. It has been reported that adenosine $A_{2A}$ agonists, e.g. the CGS 21680

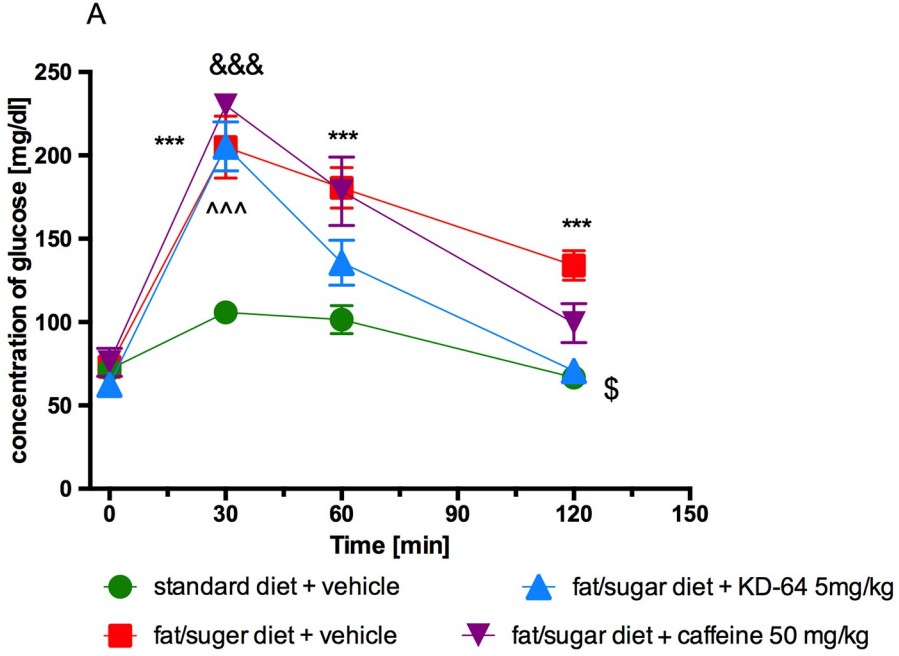

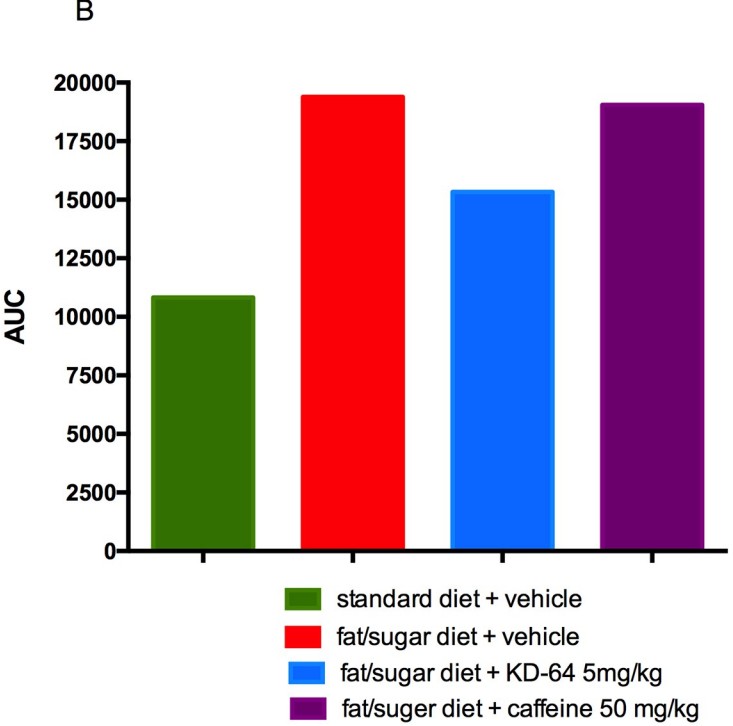

**Fig 8. Glucose tolerance test.** (**A**) Intraperitoneal glucose tolerance test (IPGTT), (**B**) area under the curve of IPGTT. Results are expressed as means ± SEM, n = 8. Comparisons were performed by Multi-t test. * Significant between control mice fed standard diet and control mice fed fat/sugar diet, ^ Significant between control mice fed standard diet and mice fed fat/sugar diet and treated with KD-64, & Significant between control mice fed standard diet and mice fed fat/sugar diet and treated with caffeine, $ Significant between control mice fed fat/sugar diet and mice fed fat/sugar diet and treated with KD-64; $p<0.05, ***,^^^,&&&p<0.001.

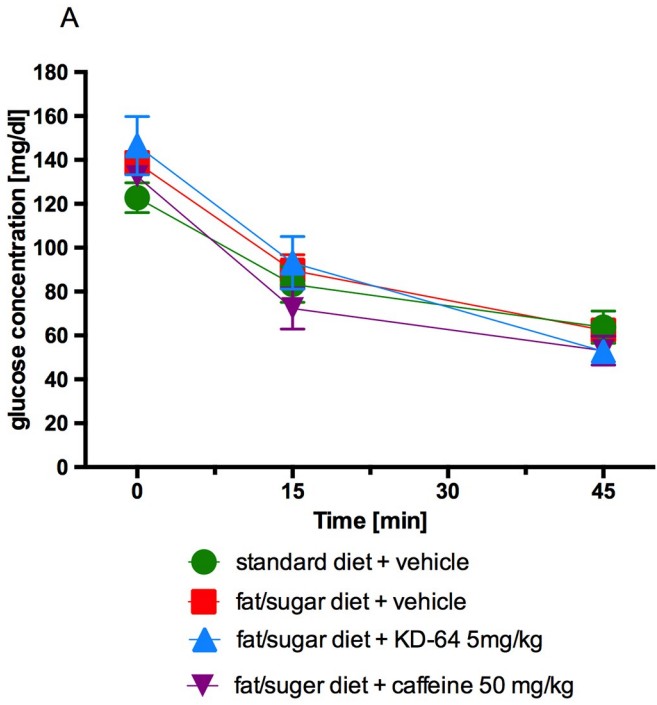

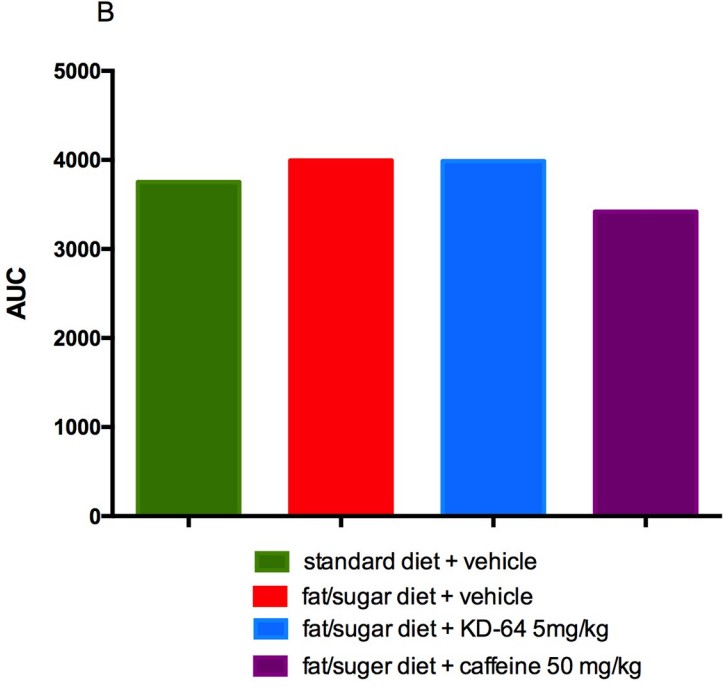

**Fig 9. Insulin sensitivity test.** (**A**) Insulin tolerance test (ITT), (**B**) area under the curve of the ITT. Results are expressed as means ± SEM, n = 8. Comparisons were performed by Multi-t test.

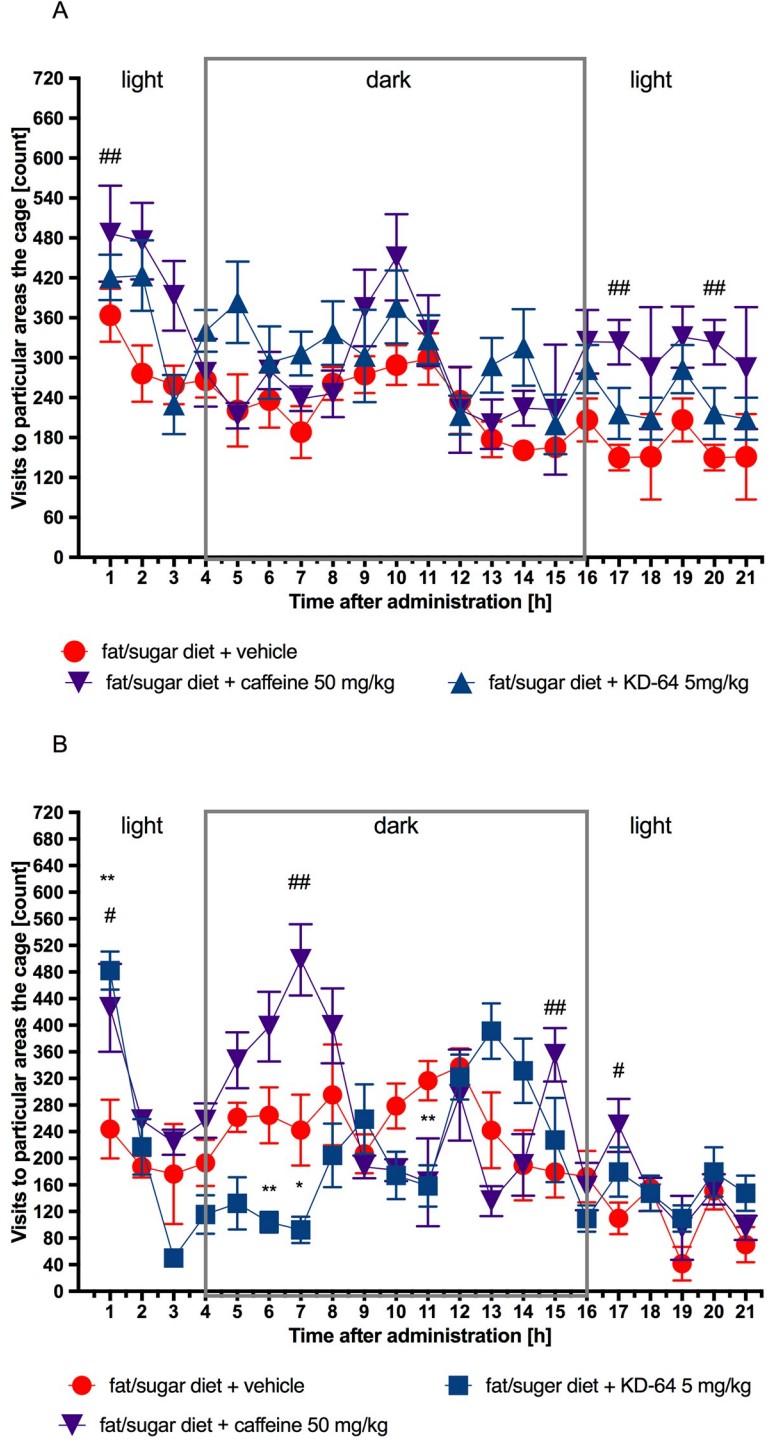

**Fig 10. Spontaneous activity after the first (A) and twentieth (B) administration of tested compounds.** Results are expressed as means ± SEM, n = 8. Comparisons were performed by Multi-t test. * Significant between control mice fed fat/sugar diet and mice fed fat/sugar diet and treated with KD-64, # Significant between control mice fed fat/sugar diet and mice fed fat/sugar diet and treated with caffeine; *,#p<0.05, **,##p<0.01.

compound, can cause sedation [56], while antagonists of this receptor, e.g. caffeine, have a stimulating effect [57].

Our research showed that the selective $A_{2A}$ receptor antagonist KD-64 did not induce changes in spontaneous activity after a single administration therefore we proceeded with the further experiments. In models of both local and systemic inflammation the anti-inflammatory effect of the KD-64 compound administered at a dose of 5 mg/kg b.w./day was comparable to the anti-inflammatory effect of ketoprofen (5 mg/kg b.w./day) used as a reference standard. There are reports in the literature that pharmacological blockade of selected adenosine receptor subtypes (PSB-36, PSB-1115, MSX-3, and PSB-10) after systemic application of antagonists generally leads to decrease in edema formation after the carrageenan injection [58]. In the case of $A_{2A}$ antagonist, activity of the drug varies with time, suggesting that the importance of adenosine receptor activation in the inflammatory process dynamically changes in the course of inflammation [58]. The local injection of the highly $A_{2A}$-selective agonist CGS21680 induced paw edema, conversely [59] $A_{2A}$ antagonist MSX-3, at a dose of 10 mg/kg b.w., significantly reduced the carrageenan-induced edema [58]. Similarly, in our study, we showed that intraperitoneal injection of selective $A_{2A}$ receptor antagonist KD-64 at a dose of 5 mg/kg b.w./day, results in a potent inhibition of carrageenan-induced edema formation (Fig 3). Thus, the results obtained are in line with literature reports and clearly show that selective $A_{2A}$ receptor antagonists have anti-inflammatory effect.

Adenosine may be added to the growing list of key signalling molecules that regulate vascular function and homeostasis and to a very selected list of agonists that promote integrity of the vascular bed [60]. There are reports in the literature that adenosine regulates the pulmonary endothelial cells barrier function via $A_{2A}$ receptors [60] and through its influence on the adenosine receptors $A_1$ and $A_{2A}$ can modulate for example blood-brain barrier permeability [61]. Blocking just the adenosine $A_{2A}$ receptor reduces permeability and blocks the entry of inflammatory cells and soluble factors into the brain [61]. 1It has been also reported that vascular permeability in the hind plantar skin of rats decreases following lumbar sympathectomy, possibly via reduction of adenosine receptor $A_{2A}$ expression [62]. In the second model of inflammation used in the presented study i.e. zymosan-induced peritonitis, KD-64 showed significant anti-inflammatory effect, manifested by decrease of both vascular permeability, and plasma neutrophils count. The results were comparable to the ones obtained after ketoprofen administration. It is an important finding, confirming that $A_{2A}$ adenosine receptor blockade may be directly responsible for a decrease in vascular permeability.

Anti-inflammatory effect of KD-64 was also evaluated through the measurement of CRP levels—an acute phase protein primarily expressed and secreted by the liver. In response to tissue injury or infection, the plasma concentrations of CRP can increase rapidly, moreover CRP level also increases in chronic inflammatory diseases, including cardiovascular and autoimmune disease. Due to the correlation between CRP and inflammation, CRP has attracted wide attention as a non-specific marker used for purpose of evaluation and monitoring of the infection and inflammation development as well as a prognostic marker for cardiovascular events [63]. In both models of inflammation, the tested antagonist of $A_{2A}$ adenosine receptors KD-64 statistically significantly reduced the level of CRP in plasma. This confirms the anti-inflammatory efficacy of this compound after its *ip* administration at a dose of 5 mg/kg b.w./day.

Based on the described above findings the dose of 5 mg/kg b.w./day of KD-64 was selected for testing in mice obesity model. During the first days of KD-64 administration, obese animals weighed significantly less compared to the obese control mice. But then they began to gain weight at the rate comparable to obese control mice. Not surprisingly, the amount of peritoneal adipose tissue, measured at the end of the experiment, was comparable in these two groups of animals. It is interesting, however, that the group receiving caffeine at a dose of 50

mg/kg/ b.w./day which gained weight significantly less than obese control mice (results consistent with the literature findings) [64] at the end of the experiment had the amount of peritoneal fat also comparable to the other experimental groups.

As the compound KD-64 was administered, its effectiveness in reducing weight of obese mice was decreasing. Unfortunately, the conducted research does not provide an assessment of why this might have happened. More detailed investigation is needed to determine if, for example, it was due to changes in the number and sensitivity of adenosine receptors with repeated administrations. Literature reports such cases that repeated administrations of adenosine A$_{2A}$ ligands may lead to the changes in regulation (both up- or down-regulation) of the A$_{2A}$ adenosine receptor gene [65]. In addition, various pathological conditions can cause changes in the density of A$_{2A}$ adenosine receptors, for example it has been shown that pro-inflammatory stimuli up-regulate A$_{2A}$ adenoside receptor and for the effective treatment appropriately higher doses of ligands are required [66]. A reduction in DNA methylation at the A$_{2A}$ adenosine receptor gene promoter site and an increase in the protein levels and gene expression in binge-like-eating rats has been also reported [65]. Interestingly, alterations of DNA methylation of A$_{2A}$ adenosine receptor has been observed in other diseases such as schizophrenia [67], Huntington's disease [68] and cardiomyopathies [69].

Studies of spontaneous activity in obese mice showed that repeated administration of KD-64 led to a decrease in spontaneous activity at some hours after the twentieth administration of this compound. As previously mentioned, adenosine A$_{2A}$ agonists, e.g. the CGS 21680 compound, can cause sedation [56], which may indicate that in fact a change in receptor density and sensitivity (up-regulation) after repeated administration of KD-64 could be the cause of such observation.

In obesity, the white adipose tissue produces large numbers of inflammatory agents including TNF-α and IL-6, which can affect the physiology of the adipose tissue locally, but also may induce systemic effects on the other organs [70]. IL-6 has emerged as one of the mediators linking obesity-derived chronic inflammation with insulin resistance. In high fat diet fed obese mice, activated hepatic IL-6 signalling is accompanied by systemic and local insulin resistance, which can be reversed by neutralization of IL-6 [52]. Indeed, in our study, the levels of TNF-α and IL-6 in the plasma of obese mice were higher as compared to control non-obese mice and were significantly reduced by the administration of KD-64. This reduction in the levels of inflammatory cytokines is probably the result of the anti-inflammatory effect of the tested compound. It should be emphasized that these tests were made with homogeneous, very sensitive and reliable methods. In addition, in the glucose load test, significant differences were observed in the response curve in group which received KD-64 treatment compared to the obese control group. The glucose level in mice treated with KD-64 one hour after loading did not differ statistically from the level determined in non-obese mice fed standard feed. This indicates an improvement in glucose tolerance in KD-64-treated mice compared to the obese control mice which might be connected to the anti-inflammatory effect of the tested compound and a decrease in plasma IL-6 levels, since IL-6 and its signalling path play complex roles in metabolic disorders [71]. It was recently reported that IL-6 enhances fatty acid synthesis in murine hepatocytes via the induction of the citrate transporter *mIndy* [72] and it also exacerbates hepatic inflammation and steatosis [73]. Additionally, adenosine has been shown to promote IL-6 production [74] and our study demonstrates that adenosine A$_{2A}$ receptor may be associated with this activity, because selective antagonist of this particular receptor decreased IL-6 level in plasma. In contrast, several human studies reported that despite the well-proven anti-inflammatory effect of caffeine [75, 76, 77], its administration, leads to an increase in serum IL-6 levels [78, 79]. The results of our study, even though performed on mice, are in line with these observations. Observed in our studies differences in plasma IL-6

levels between the caffeine and KD-64 treatment groups may indicate that decrease in IL-6 concentration is due not only to the anti-inflammatory effect of the KD-64 compound, but also adenosine A$_{2A}$ receptor blockade. This is an interesting topic that undoubtedly requires further research.

Another interesting and very favourable result of KD-64 treatment is its ability to normalize the elevated AlaT activity, induced by high-fat/sugar feeding. Liver is a vital organ involved in detoxification and drug metabolism, therefore potential hepatotoxicity could eliminate the test compound from further stages of development especially if it is intended for longer use. In our experiment caffeine did not have any effect on AlaT levels, although there are reports in the literature that in the obesity model caused by the administration of high-fat feed, caffeine administered for several weeks at the dose of 20–40 mg/kg in the drinking water normalizes liver enzymes [64]. Probably this effect depends also on the dose and time of caffeine administration.

In conclusion, after repeated administrations of a selective A$_{2A}$ adenosine receptor antagonist compound KD-64 with documented anti-inflammatory activity, no significant reduction in weight gain was observed in obese mice fed high-calorie feed. However, contrary to caffeine (non-selective adenosine receptor antagonist) investigated compound normalized levels of selected cytokines and inflammatory proteins, including IL-6, as well as, activity of alanine transaminase and improved glucose tolerance in the obese mice. Thus our findings prove that the activity of the selective adenosine A$_{2A}$ receptor antagonist is different from that of the non-selective antagonist.

## Acknowledgments

The authors wish to gratefully acknowledge Maria Kaleta and Joanna Knutelska for their technical assistance.

## Author Contributions

**Conceptualization:** Magdalena Kotańska.

**Formal analysis:** Magdalena Kotańska.

**Funding acquisition:** Magdalena Kotańska, Anna Drabczyńska, Katarzyna Kieć-Kononowicz.

**Investigation:** Magdalena Kotańska, Anna Dziubina, Karolina Reguła, Marek Bednarski, Małgorzata Zygmunt.

**Methodology:** Magdalena Kotańska.

**Project administration:** Magdalena Kotańska.

**Resources:** Magdalena Kotańska, Anna Dziubina, Kamil Mika, Karolina Reguła, Anna Drabczyńska, Katarzyna Kieć-Kononowicz.

**Supervision:** Magdalena Kotańska, Małgorzata Szafarz, Jacek Sapa, Katarzyna Kieć-Kononowicz.

**Visualization:** Magdalena Kotańska, Anna Dziubina, Małgorzata Szafarz, Karolina Reguła, Marek Bednarski.

**Writing – original draft:** Magdalena Kotańska, Anna Dziubina, Małgorzata Szafarz, Kamil Mika, Marek Bednarski.

**Writing – review & editing:** Magdalena Kotańska, Anna Dziubina, Małgorzata Szafarz, Kamil Mika, Jacek Sapa, Katarzyna Kieć-Kononowicz.

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
