## [Decision Letter · Decision Letter 0]

17 Mar 2020

PONE-D-20-04266

KD-64 – a new selective A2A adenosine receptor antagonist has anti-inflammatory activity but contrary to the non-selective antagonist – caffeine does not reduce diet-induced obesity in mice

PLOS ONE

Dear Dr Kotańska,

Thank you for submitting your manuscript to PLOS ONE. After careful consideration, we feel that it has merit but does not fully meet PLOS ONE’s publication criteria as it currently stands. Therefore, we invite you to submit a revised version of the manuscript that addresses the points raised during the review process.

We would appreciate receiving your revised manuscript by May 01 2020 11:59PM. To enhance the reproducibility of your results, we recommend that if applicable you deposit your laboratory protocols in protocols.io, where a protocol can be assigned its own identifier (DOI) such that it can be cited independently in the future. For instructions see: http://journals.plos.org/plosone/s/submission-guidelines#loc-laboratory-protocols

We look forward to receiving your revised manuscript.

Kind regards,

Marco Aurélio Gouveia Alves

Academic Editor

PLOS ONE

Journal Requirements:

2. At this time, we request that you  please report additional details in your Methods section regarding animal care,: a) Please include the source of the animals used in this study and  b) please describe any steps taken to minimize animal suffering and distress, such as by administering anesthetics or analgesics.

Reviewers' comments:

Reviewer's Responses to Questions

**Comments to the Author**

1. Is the manuscript technically sound, and do the data support the conclusions?

Reviewer #1: Partly

Reviewer #2: Yes

2. Has the statistical analysis been performed appropriately and rigorously? 

Reviewer #1: Yes

Reviewer #2: Yes

3. Have the authors made all data underlying the findings in their manuscript fully available?

Reviewer #1: Yes

Reviewer #2: Yes

4. Is the manuscript presented in an intelligible fashion and written in standard English?

Reviewer #1: Yes

Reviewer #2: Yes

5. Review Comments to the Author

Reviewer #1: General Comments:

According to the World Health Organization (WHO), obesity has reached epidemic proportions globally. Obesity causes low-grade chronic inflammation which may lead to a number of chronic diseases. Additionally, A2 adenosine receptors play an important role in the regulation of the inflammatory process and glucose homeostasis.

Although caffeine, a non-selective A2A adenosine receptor antagonist, has been extensively studied by some research groups, the anti-inflammatory and anti-obesity mechanisms of some selective A2A adenosine receptor antagonists are few studied.

This paper reports the anti-inflammatory and anti-obesity activities of KD-64, a new selective A2A adenosine receptor antagonist, by using several current techniques and animal models.

The paper is original, well written and organized, and within the scope of Plos One.

Specific Comments

Title

The title should be concise. Authors should consider to reduce it.

Introduction

In page 5 – lines 106 and 107, it is referred that “…These compounds have similar structure to KD-64 – ligand we have used in the presented study…”.

KD-64 IUPAC chemical name must be included and the similarities between the chemical structures of KD-64 and the other referred compounds should be described.

Is KD-64 a caffeine analogue?

Experimental Design / Results and Discussion

Concerning the three animal models, more details must be included, namely:

• The age of the animals at the beginning of the experiences

• Considering that gender is a confounding variable, wouldn’t be better to use only one gender in all groups/models of the study? (Usually, male mice provide more pure and accurate responses…)

• The concentration of caffeine used was much higher than the concentration of KD-64. Why? (This may explain, at least partially, the reported caffeine pro-inflammatory effect).

• Standard diet composition should be provided in a table (together with the high fat-diet composition)

Reviewer #2: Dear Kotanska and colleagues,

I have some minor questions regarding your article:

- line 73-75 "the role of adenosine receptor signaling in the development and progressiona of numerous diseases has been emphasized for years" needs a reference.

- line 146: "in all experiments ketoprofenm caffeine or KD-64 were administered as suspensions in 1% Tween 80" in what solution? PBS? NaCL?

- line 258: how did you measure "spontanous activity"?

- figure 4 C in yy axis please add the wavelenght the absorbance was measured

6. PLOS authors have the option to publish the peer review history of their article (what does this mean?). If published, this will include your full peer review and any attached files.

Reviewer #1: No

Reviewer #2: No

---

## [Author Response · Author response to Decision Letter 0]

8 Apr 2020

Journal Requirements:

Manuscript meets PLOS ONE's style requirements, including those for file naming.

 2. At this time, we request that you please report additional details in your Methods section regarding animal care,: a) Please include the source of the animals used in this study and  b) please describe any steps taken to minimize animal suffering and distress, such as by administering anesthetics or analgesics.

Source of the animals used in this study has been included. LINES: 138-140

Following sentence has been added: “The animals were obtained from the Animal House of the Faculty of Pharmacy of the Jagiellonian University Medical College.

Steps taken to minimize animal suffering and distress have been described. LINES: 147-155

Following paragraph has been added:

“During the experiments, when possible, steps were taken to minimize animal suffering and distress. Unfortunately, in the anti-inflammatory studies, it was not possible to administer any anaesthetics and analgesics since they would interfere with the tests, however the animals were not isolated individually to avoid developing stress. In the studies of anti-obesity properties efforts were made to create the safe and stress-free conditions for animals so that stress would not affect the results of experiments. During the induction of obesity, only two people responsible for the subsequent administration of the investigated compounds, were allowed to handle the animals. After experiment mice were killed by decapitation that is considered to be the most humane method of mouse euthanasia”.

3. We note that you have included the phrase “data not shown” in your manuscript. Unfortunately, this does not meet our data sharing requirements. PLOS does not permit references to inaccessible data. We require that authors provide all relevant data within the paper, Supporting Information files, or in an acceptable, public repository. Please add a citation to support this phrase or upload the data that corresponds with these findings to a stable repository (such as Figshare or Dryad) and provide and URLs, DOIs, or accession numbers that may be used to access these data. Or, if the data are not a core part of the research being presented in your study, we ask that you remove the phrase that refers to these data. 

These data are not a core part of research being presented in our study. The phrase has been deleted. LINE 179

Reviewers' comments:

5. Review Comments to the Author  Please use the space provided to explain your answers to the questions above. You may also include additional comments for the author, including concerns about dual publication, research ethics, or publication ethics. (Please upload your review as an attachment if it exceeds 20,000 characters)  

Reviewer #1: General Comments:  According to the World Health Organization (WHO), obesity has reached epidemic proportions globally. Obesity causes low-grade chronic inflammation which may lead to a number of chronic diseases. Additionally, A2 adenosine receptors play an important role in the regulation of the inflammatory process and glucose homeostasis. Although caffeine, a non-selective A2A adenosine receptor antagonist, has been extensively studied by some research groups, the anti-inflammatory and anti-obesity mechanisms of some selective A2A adenosine receptor antagonists are few studied. This paper reports the anti-inflammatory and anti-obesity activities of KD-64, a new selective A2A adenosine receptor antagonist, by using several current techniques and animal models. The paper is original, well written and organized, and within the scope of Plos One.

   

Specific Comments  

Title The title should be concise. Authors should consider to reduce it.

We would like to leave it the way it is. In our opinion it contains the most interesting points of our investigation and in this way it might draw the attention of a potential reader.   

Introduction In page 5 – lines 106 and 107, it is referred that “…These compounds have similar structure to KD-64 – ligand we have used in the presented study…”. KD-64 IUPAC chemical name must be included and the similarities between the chemical structures of KD-64 and the other referred compounds should be described. Is KD-64 a caffeine analogue?

Introduction has been supplemented with the appropriate information. LINES 102-103, 106-110, 111, 131-132, 161-162, Figure 1

  Experimental Design / Results and Discussion 

Concerning the three animal models, more details must be included, namely: • The age of the animals at the beginning of the experiences

Six-week old mice were used for both acute and obesity induction models. These information was added to manuscript in the Methods section. LINES 136, 137

 • Considering that gender is a confounding variable, wouldn’t be better to use only one gender in all groups/models of the study? (Usually, male mice provide more pure and accurate responses…)

We agree that male mice provide more accurate responses therefore male mice were used in the acute models. However, in the obesity model, female mice were used for a specific reason.

Induction of obesity is difficult. It is desirable to get the largest differences between the weight of lean mice - standard control and the weight of obese mice - obese control. During induction of obesity the body weight of female mice stabilizes faster than that of males and in the following weeks females no longer gain weight. Males grow larger so their body weight changes longer. Since control female mice change little weight, while control obese mice quickly increase weight, in the model of obesity induced by the administration of high-fat feed a greater weight difference between these two groups can be obtained. When the differences in body weight are greater, the pharmacological effects of the tested compounds can be seen better.

 • The concentration of caffeine used was much higher than the concentration of KD-64. Why? (This may explain, at least partially, the reported caffeine pro-inflammatory effect).

The caffeine dose of 50 mg/kg b.w./day, i.p. was selected based on the literature. 

In studies on the effects of caffeine on body weight of obese mice, Wu et al., 2017 administered 60 mg/kg b.w./day, p.o. In studies on its analgesic effect in mice, Bach-Rojecky, 2003 used doses from 1.67 to 67 mg/kg b.w., i.p. Caffeine exerts a direct dose-dependent analgesic action. Pohanka, 2014, showed antioxidant activity of caffeine at the doses from 1 – 67 mg/kg b.w., i.m. Other study shows that, administration of a dose of 25 mg/kg b.w. to rats led to its anti-inflammatory effect [Li et al., 2011]. Caffeine at a dose of 100 mg/kg b.w. provided strong protection against acute liver damage, but at a dose of 25 mg/kg b.w. exacerbated liver damage. Ohta et al., 2007 showed that the effect of caffeine on inflammation is biphasically dependent on doses. Namely, lower doses of caffeine enhanced proinflammatory cytokine induction and tissue destruction, while a high dose of caffeine suppressed tissue damage by the inhibition of proinflammatory cytokine responses and the induction of an anti-inflammatory cytokine, IL-10. 

As we wrote in the manuscript, caffeine administration has led to an increase in plasma IL-6 levels in some studies [Walker et al., 2007, Tauler et al., 2013]. In our study the same effect was observed. This does not mean, however, that caffeine has a pro-inflammatory effect.

Therefore it is difficult to answer the question whether the dose of 50 mg/kg b.w. could lead to the pro-inflammatory effects of caffeine.

Wu L, Meng J, Shen Q, Zhang Y, Pan S, Chen Z, Zhu LQ, Lu Y, Huang Y, Zhang G. Caffeine inhibits hypothalamic A1R to excite oxytocin neuron and ameliorate dietary obesity in mice. Nat Commun. 2017 Jun 27;8:15904. doi: 10.1038/ncomms15904.

Bach-Rojecky L. Analgesic effect of caffeine and clomipramine: a possible interaction between adenosine and serotonin systems. Acta Pharm. 2003 Mar;53(1):33-9.

Pohanka M. Caffeine alters oxidative homeostasis in the body of BALB/c mice. Bratisl Lek Listy. 2014;115(11):699-703.

Li XY, Xu L, Lin GS, Li XY, Jiang XJ, Wang T, Lü JJ, Zeng B. Protective effect of caffeine administration on myocardial ischemia/reperfusion injury in rats. Shock. 2011 Sep;36(3):289-94. doi: 10.1097/SHK.0b013e3182222915.

Ohta A, Lukashev D, Jackson EK, Fredholm BB, Sitkovsky M. 1,3,7-trimethylxanthine (caffeine) may exacerbate acute inflammatory liver injury by weakening the physiological immunosuppressive mechanism. J Immunol. 2007 Dec 1;179(11):7431-8.

Walker GJ, Finlay O, Griffiths H, Sylvester J, Williams M, Bishop NC. Immunoendocrine response to cycling following ingestion of caffeine and carbohydrate. Med Sci Sports Exerc. 2007;39:1554-1560.

Tauler P, Martinez S, Moreno S, Monjo M, Martínez P, Aguiló A. Effects of caffeine on the inflammatory response induced by a 15-km run competition. Med Sci Sports Exerc. 2013;45:1269–1276.

 • Standard diet composition should be provided in a table (together with the high fat-diet composition)

Appropriate information has been added. LINES: 241-249  

Reviewer #2: Dear Kotanska and colleagues,

I have some minor questions regarding your article: - line 73-75 "the role of adenosine receptor signaling in the development and progressiona of numerous diseases has been emphasized for years" needs a reference.

The reference has been added and all subsequent literature renumbered. LINE 75, 610-621

Chen JF, Eltzschig HK, Fredholm BB. Adenosine receptors as drug targets--what are the challenges? Nat Rev Drug Discov. 2013 Apr;12(4):265-86. doi: 10.1038/nrd3955. Review.

 - line 146: "in all experiments ketoprofenm caffeine or KD-64 were administered as suspensions in 1% Tween 80" in what solution? PBS? NaCL?

These compounds were administered in solution of 1% Tween 80 in sterile water for injection. Appropriate information has been added. LINE 166 -167

 - line 258: how did you measure "spontanous activity"?

In this place of the manuscript I should have used wording „locomotor activity”. The sentence has been corrected. LINE 297

 - figure 4 C in yy axis please add the wavelenght the absorbance was measured

The wavelength (620nm) has been added.

---

## [Decision Letter · Decision Letter 1]

29 May 2020

KD-64 – a new selective A2A adenosine receptor antagonist has anti-inflammatory activity but contrary to the non-selective antagonist – caffeine does not reduce diet-induced obesity in mice

PONE-D-20-04266R1

Dear Dr. Kotańska,

We are pleased to inform you that your manuscript has been judged scientifically suitable for publication and will be formally accepted for publication once it complies with all outstanding technical requirements.

With kind regards,

Marco Aurélio Gouveia Alves

Academic Editor

PLOS ONE

Additional Editor Comments (optional):

Reviewers' comments:

Reviewer's Responses to Questions

**Comments to the Author**

1. If the authors have adequately addressed your comments raised in a previous round of review and you feel that this manuscript is now acceptable for publication, you may indicate that here to bypass the “Comments to the Author” section, enter your conflict of interest statement in the “Confidential to Editor” section, and submit your "Accept" recommendation.

Reviewer #2: All comments have been addressed

Reviewer #3: All comments have been addressed

2. Is the manuscript technically sound, and do the data support the conclusions?

Reviewer #2: Yes

Reviewer #3: Yes

3. Has the statistical analysis been performed appropriately and rigorously? 

Reviewer #2: Yes

Reviewer #3: Yes

4. Have the authors made all data underlying the findings in their manuscript fully available?

Reviewer #2: Yes

Reviewer #3: Yes

5. Is the manuscript presented in an intelligible fashion and written in standard English?

Reviewer #2: Yes

Reviewer #3: Yes

6. Review Comments to the Author

Reviewer #2: Dear authors,

Thank you for revising the manuscript accordingly with reviewers instructions. It is now suitable for publication.

Reviewer #3: (No Response)

7. PLOS authors have the option to publish the peer review history of their article (what does this mean?). If published, this will include your full peer review and any attached files.

Reviewer #2: No

Reviewer #3: No

---

## [Editor Report · Acceptance letter]

3 Jun 2020

PONE-D-20-04266R1 

KD-64 – a new selective A2A adenosine receptor antagonist has anti-inflammatory activity but contrary to the non-selective antagonist – caffeine does not reduce diet-induced obesity in mice 

Dear Dr. Kotańska:

I'm pleased to inform you that your manuscript has been deemed suitable for publication in PLOS ONE. Congratulations! Your manuscript is now with our production department. 

Kind regards, 

on behalf of

Dr. Marco Aurélio Gouveia Alves 

Academic Editor

PLOS ONE